# Autonomous Interactive Correction MLLM for Robust Robotic Manipulation

**Chuyan Xiong**[1,2*]  **Chengyu Shen**[1*]  **Xiaoqi Li**[1*]  **Kaichen Zhou**[1†]
**Jeremy Liu**[1]  **Ruiping Wang**[2]  **Hao Dong**[1‡]
[1]School of Computer Science, Peking University
[2]Institute of Computing Technology, Chinese Academy of Sciences

**Abstract:** The ability to reflect on and correct failures is crucial for robotic systems to interact stably with real-life objects. Observing the generalization and reasoning capabilities of Multimodal Large Language Models (MLLMs), previous approaches have aimed to utilize these models to enhance robotic systems accordingly. However, these methods typically focus on high-level planning corrections using an additional MLLM, with limited utilization of failed samples to correct low-level contact poses which is particularly prone to occur during articulated object manipulation. To address this gap, we propose an Autonomous Interactive Correction (AIC) MLLM, which makes use of previous low-level interaction experiences to correct SE(3) pose predictions for articulated object. Specifically, AIC MLLM is initially fine-tuned to acquire both pose prediction and feedback prompt comprehension abilities. We design two types of prompt instructions for interactions with objects: **1)** visual masks to highlight unmovable parts for position correction, and **2)** textual descriptions to indicate potential directions for rotation correction. During inference, a Feedback Information Extraction module is introduced to recognize the failure cause, allowing AIC MLLM to adaptively correct the pose prediction using the corresponding prompts. To further enhance manipulation stability, we devise a Test Time Adaptation strategy that enables AIC MLLM to better adapt to the current scene configuration. Finally, extensive experiments are conducted in both simulated and real-world environments to evaluate the proposed method. The results demonstrate that our AIC MLLM can efficiently correct failure samples by leveraging interaction experience prompts. Our project website is https://sites.google.com/view/aic-mllm.

**Keywords:** Robot Manipulation, Large Language Model, Failure Correction

## 1 Introduction

Developing a versatile robot has always been a core goal in embodied artificial intelligence. Despite various generalization strategies proposed [1, 2, 3, 4], failures remain inevitable due to the complexity of real-world environments. Therefore, beyond directly enhancing the robot's generalization ability, it is also crucial to enable the robot to reflect on and correct its failure actions [5, 6, 7].

In recent years, the powerful common-sense generalization and reasoning abilities of Multimodal Large Language Models (MLLMs) [8, 9, 10, 11, 12, 13] have attracted the attention of robotic researchers. These capabilities make MLLMs and LLMs naturally suited for understanding, analyzing, explaining failures, and even making corrections based on them. REFLECT [6] innovatively summarizes hierarchical sensor information and inputs it into an LLM in textual form to obtain explanations for errors. Additionally, other works [14, 15, 16, 7, 17, 18, 19, 20] are also exploring the ways and capabilities of LLMs in handling errors in the field of robotics.

---

[*] Equal contribution
[†] Project Leader
[‡] Corresponding author

8th Conference on Robot Learning (CoRL 2024), Munich, Germany.

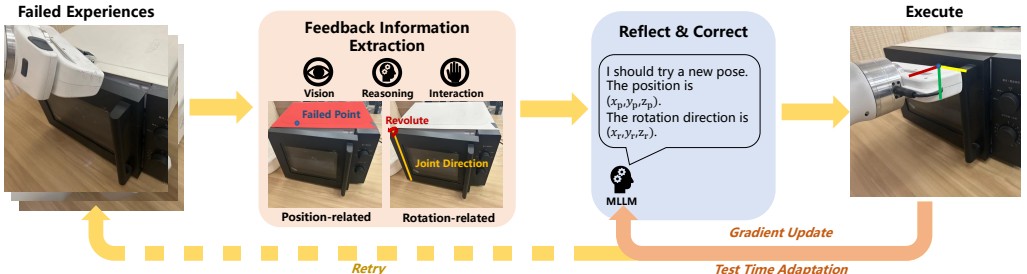

Figure 1: **Correction process of AIC MLLM.** Given a failed interaction, we first extract feedback information regarding the object's geometry, then enable the model to reflect and correct both position and rotation estimation, thereby generating a more accurate SE(3) pose which will be executed by robot. We use a right-hand coordinate system to show the end effector's rotation pose. In the rightmost image, the yellow line represents the x-axis, the red line represents the y-axis, and the red line represents the z-axis.

While these works can reflect and correct high-level tasks, such as task planning, they still struggle to correct low-level SE(3) pose actions and still fail to complete the atomic tasks decomposed by task planning. For example, even if a robot recognizes that it failed to open a microwave to place food inside, it may still encounter difficulty in selecting the correct contact pose, ultimately resulting in task failure. In such cases, previous methods cannot automatically recover from these errors [6]. To address this gap, we focus on low-level pose correction for atomic tasks. Among these tasks, articulated objects require more precise interaction poses due to their multiple interconnected parts that can move relative to each other, leading to higher degrees of freedom and increased complexity in manipulation. As a result, tasks involving articulated objects are more prone to errors and urgently need effective correction mechanisms. Therefore, we aim to develop the correction of articulated object manipulation. Given that learning from failed scenarios is an inherent capability of humans, we aim to enhance the robot's ability to understand geometry by utilizing failed attempts, and thus improve its generation of low-level actions.

Leveraging this insight, we propose an Autonomous Interactive Correction (AIC) MLLM, which utilizes previous low-level interaction experiences to correct manipulation SE(3) poses. First, we fine-tune a pre-trained MLLM to enable it not only to predict manipulation poses but also to understand the instructions provided by the feedback information extraction system and reflect on them for correction. To realize this goal, we carefully design two types of prompt instructions to assist the MLLM in failure correction, including position and rotation prompts. Specifically, we create visual masks to highlight unmovable parts for position correction and use textual descriptions to indicate potential manipulation directions for rotation correction. During inference, the model initially performs pose prediction. If an error occurs, the Feedback Information Extraction module is introduced to detect the cause of failure, enabling the AIC MLLM to adaptively correct the end-effector pose prediction using the corresponding prompts. To further enhance manipulation stability, we devise a Test Time Adaptation strategy for AIC MLLM. This allows our model to iteratively update itself during inference and adapt to the current scene configuration. With our proposed AIC MLLM, the robot can efficiently improve its manipulation stability through interactive attempts.

The main contributions of this paper can be summarized as follows: 1) We introduce AIC MLLM, a framework that utilizes MLLM for correcting SE(3) pose predictions by learning from low-level interaction failures. 2) We design visual and textual prompts to guide position and rotation corrections, and incorporate a feedback information extraction module to adaptively correct pose predictions based on identified failure causes. 3) We implement a test-time adaptation module to enhance manipulation stability. Extensive experiments demonstrate the efficacy of AIC MLLM in both simulated and real-world environments.

## 2   Related Work

**Robotic Failure Correction.** As Multimodal Large Language Models (MLLMs) advance, robot correction mechanisms have garnered increasing attention. These works fall into two main cate-

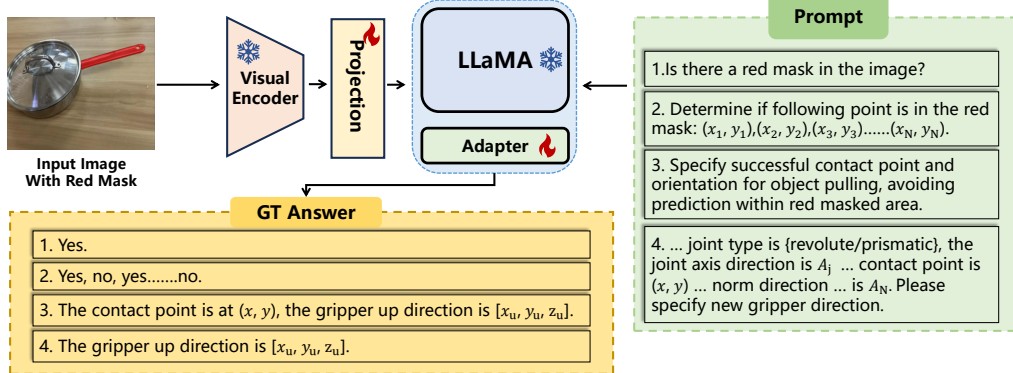

Figure 2: **Training of AIC MLLM.** We gradually enable the model to predict poses and comprehend both visual and textual feedback prompts including object parts and axis information.

gories: human-guided correction and self-correction. the former approaches [17, 7, 15] like YAY robot [17] and DROC [7] utilize human language correction to adjust robot behavior. Conversely, self-reflective methods [18, 14, 6, 16, 19, 20] such as inner monologue [14] and REFLECT [6] enable robots to learn from errors and correct autonomously. We focus on the latter type, aiming to develop a system capable of automatic self-reflection and correction at low-level pose stage. Due to space limitations, we provide additional related work in Appendix A.

## 3 Method

### 3.1 Task Formulation

We define the manipulation task as follows: the policy model $\pi$ predicts a manipulation contact pose $a(a^{pos}, a^{rot})$ based on the current scene image $I$ and the instruction $L$, which includes the required action primitive for the task. After contacting the object, we use the axis direction of end-effector as the pose trajectory after contacting (for pull primitive we use red line y-axis in Fig. 1) . The model $\pi$ is allowed $N$ opportunities to correct $a$, and during the testing phase, the model can update itself using only the current sample.

### 3.2 Framework Overview

During training (3.3), we enable the model to predict poses, and comprehend given feedback prompts, such as visual masks highlighting unmovable part or text describing axis information, and correct the pose prediction accordingly. During testing, if interaction fails, we use an **F**eedback **I**nformation **E**xtraction (**FIE**) module to obtain error information related to the geometry articulated object (3.4.1). This error information is then fed back into the trained model, allowing it to reflect on the failure's cause and predict a new SE(3) pose for interacting with the object, aiming to achieve successful manipulation (3.4.2). To enable the model to better adapt to the current testing configuration, we devise a Test Time Adaptation (TTA) strategy. This strategy allows the model to learn from the ongoing sample and improve its ability to handle upcoming samples (3.4.3).

### 3.3 Training Phase

In this section, we introduce how we train our model capable of both executing manipulation tasks and correcting failure interaction based on feedback. To achieve this goal, we first enable our model to output the manipulation pose which contains both position and rotation, enabling interaction with the object. Next, to enable our model to correct failure interaction based on feedback, we have constructed two prompt interfaces: a visual mask prompt and linguistic prompts. These interfaces convey part positional and rotational information of the object, extracted from failure samples, to the model. This enables the model to generate new predictions based on this prior knowledge.

### 3.3.1 Positional Error Interface

Based on our observations, if the model predicts points on unmovable parts, no method can successfully manipulate these parts. Therefore, to prevent the model from making invalid interactions on unmovable parts, we summarize these positional error information by segmenting out unmovable part, and enable our model to comprehend it.

Specifically, we denote a part of the articulated object as unmovable by applying a red-colored mask. We then ensure that our model comprehends the red mask on image and avoid from predicting on unmovable parts through progressively training with the following three VQA tasks:

**(1) Mask Image Classification** To ensure our model recognizes unmovable parts labeled with a red mask, we first train it to detect whether a mask is present in the image. We add a red mask to the unmovable parts in the simulator and prompt the model with the question, "Is there a red mask in the image?" The ground truth answers are "Yes" or "No," supervised using cross-entropy loss $\mathcal{L}_H$.

**(2) Mask Position Reasoning** After the model learns what a mask is, we need it to identify its location. We randomly select N pixel points $(x_1, y_1)...(x_N, y_N)$ and ask the model to determine whether each point is on the mask. As shown in the second prompt on Fig.2, we format the text prompt with corresponding answer of "Yes" and "No" based on the ground truth mask, supervised using cross-entropy loss $\mathcal{L}_P$.

**(3) Correct Based On Mask** In the third prompt of Fig.2, once the model comprehends the mask and its location, we prompt it to predict a new pose based on the constraint of unmovable part. Given that MLLM are good at handling discretized data, we discretize the direction vectors into 100 bins, with each bin spanning 0.02. This task is supervised under cross-entropy loss $\mathcal{L}_C$.

### 3.3.2 Rotational Error Interface

Position-related information only affects the correction of the contact point position, while pose correction also requires adjusting the gripper direction. For articulated objects, the most crucial information for determining a valid gripper direction is the type and direction of the joint. There are primarily two types of joints: prismatic and revolute. We observe that often, even when the part of the articulated object fails to reach the expected state, it still exhibits some degree of movement. For instance, when a robotic arm tries to open a door but fails to fully complete it, external forces may still cause the door to move to some extent. Therefore, we utilize the movement in this interaction to capture information about the type of joint and the axis direction, enabling the model to reflect on and correct its actions accordingly.

Specifically, as shown in the last prompt in Fig 2, we incorporate the joint type information and direction information into the text prompt, modeling it as a VQA task. The model is required to provide new direction predictions based on joint information, the current contact point, and the current normal direction, which can also provide some hints for contact direction prediction. This task is supervised under cross-entropy loss $\mathcal{L}_R$. Once the training data is obtained from the simulator, we inject noise to simulate test conditions and improve robustness. See Appendix B.1 for details.

We jointly train the four tasks under the objective $\mathcal{L} = \mathcal{L}_H + \mathcal{L}_P + \mathcal{L}_C + \mathcal{L}_R$.

### 3.4 Testing Phase

We introduce the mechanism our framework employs during test testing phase. Specifically, given an object image, the model predicts the 2D contact point and rotation. The contact point is subsequently transformed into 3D coordinates using a depth map, enabling initial interaction with the object. If the interaction fails, we use an FIE module(3.4.1) to extract feedback information from previous failure attempt. This feedback is integrated into visual and linguistic prompts fed into the trained model, allowing it to reflect and correct, and generate new action predictions(3.4.2). Additionally, after inference on each test sample, the model undergoes parameter updates in the TTA module(3.4.3) to better generalize to configuration. The complete prompts in Fig. 3 are shown in Appendix D.

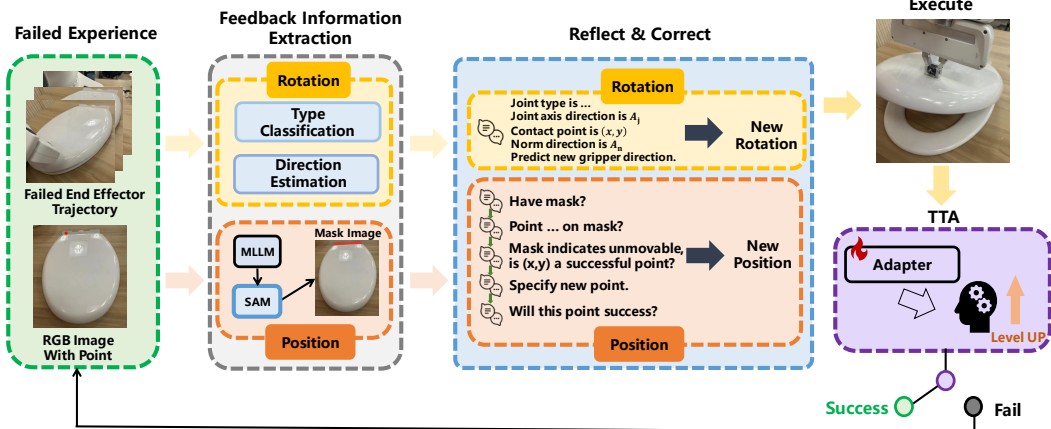

Figure 3: **Testing of AIC MLLM.** If failure interaction occurs, an FIE module is utilized to extract feedback information from previous failure attempts. This feedback information is integrated into visual and linguistic prompts, which are then fed into the trained model, enabling it to reflect, correct, and generate new action predictions. After inference on each test sample, the model undergoes parameter updates in the TTA module to enhance generalization to the current testing configuration.

### 3.4.1 Feedback Information Extraction (FIE)

During training, essential information like joint types is obtained from the simulation environment. However, during testing, these data need to be extracted automatically through specific mechanisms. This section will describe the module designed to accomplish this task.

**Position-related Information Extraction** During interaction, our goal is to determine if the predicted contact point lies on unmovable part. If so, our aim is to pinpoint the unmovable part's location and prevent from any subsequent operations on it. Specifically, we first employ the off-the-shelf VIP-LLaVA [21] to confirm whether it is unmovable. This model allows users to mark images with visual prompts, *i.e.*, a dot on the image, and interact with the model via language queries. Therefore, we draw the predicted contact point as a red dot on the object. Then, as shown in Fig. 3, we input this image with the textual prompt to VIP-LLaVA, obtaining its response to determine if the part is movable. If contact point is on unmovable part, we then use SAM [22] to perform hierarchical segmentation with the red dot as the prompt, selecting the smallest mask to represent the immovable part. Note that multiple failed interactions with an object may occur, so the unmovable parts information generated from each interaction will accumulate as multiple red masks on the object.

**Rotation-related Information Extraction** Our goal is to obtain joint type and axis direction based on the history interaction attempt. When interacting with the object, *e.g*, when pulling a door, we can determine the trajectory of the end effector. If the trajectory is a straight line, we classify it as a prismatic axis. Conversely, if the trajectory is of an arc shape, we classify it as a revolute axis. To determine the axis direction, for prismatic axis, we consider the axis direction to be the same as its direction of movement. For a revolute axis, even if the interaction fails, as long as the end effector moves, we can determine the axis direction from the end effector's trajectory. Specifically, if there is only one failed interaction, we select the start position $p_s^1$, end position $p_e^1$, and midpoint $p_m^1$ of the trajectory, all belonging to $\mathbb{R}^3$, forming two directional vectors $\overrightarrow{p_s^1 p_m^1}$ and $\overrightarrow{p_m^1 p_e^1}$. The cross product of these two vectors yields the axis direction: $\overrightarrow{A} = \overrightarrow{p_s^1 p_m^1} \times \overrightarrow{p_m^1 p_e^1}$. If there are multiple $n$ failed interactions with movement, we use the end effector's movement vectors from these interactions to perform cross products, providing a more accurate axis direction: $\overrightarrow{A} = \overrightarrow{p_s^1 p_e^1} \times ... \overrightarrow{p_s^n p_e^n}$.

### 3.4.2 Model Inference

After extracting feedback information from the failed experiences, the trained model should reflect and correct itself. Since failures may stem from either invalid position or direction, we first identify the cause before making corrections. Specifically, a failure interaction can result in either movement or no movement. In cases of the former, it indicates that the contact point is on a movable part

and can cause movement. Therefore, we consider this a valid contact point and focus on rotation correction. For the latter cause, we utilize the normal direction to interact with the object, which usually results in slight movements if the position is movable. If even the normal direction fails, we conclude that position correction is necessary.

**Position Correction** After determining position needs to be corrected, we employ chain-of-thought [23] to guide the model to understand feedback, *i.e.*, visual mask prompts, and generate new contact point to prevent from predicting on unmovable part. As shown in Fig 3, the first two steps are consistent with the training phase steps. In the third step, we prompt the model to determine whether the previously predicted points are on unmovable parts, compelling it to reflect on the reason behind the previous failures. Next, in the fourth step, we request the model to output a new SE(3) pose based on the reflection. Finally, in the fifth step, we ask the model to evaluate whether the newly predicted pose will result in a successful interaction, reinforcing its ability to reflect on the outcome.

**Rotation Correction** If rotation correction is necessary, as shown in Fig.3, we adopt the same direction correction process as in the training phase. We directly perform direction correction using the error information obtained from FIE with text prompts guidance. During this process, the contact point remains fixed and the model will output a new gripper direction.

### 3.4.3 Test Time Adaptation(TTA)

To enable continuous evolution of the model and allow it to rapidly adapt to the current configuration, we propose the Test Time Adaptation (TTA) strategy. During TTA, the model updates itself after each sample inference to learn from the sample it has just processed. The updated model then infers on the next sample, updates again, and this process of model updating continues. Specifically, we use model $\pi_{t-1}$ at timestamp $t-1$ to inference on test sample at timestamp $t$. The model can **only** utilize the attempts made on the sample at timestamp $t$ to update itself to $\pi_t$, thus enhancing its performance when dealing with subsequent samples under the same testing configuration.

The input-ground truth pairs used to update the model are as follows: To start with, since we can obtain the ground truth for the first two steps in Fig. 2—whether there is a mask and where the mask is, we update the model with these position-related VQA pairs. This enables the model to recognize which parts of the object are unmovable, thereby avoiding predictions on those parts. In addition, we also want the model to improve its manipulation ability by learning from successful correction experiences. Therefore, for poses that result in successful manipulation after correction, we use the fourth step of Fig. 2 to update the model, with the corrected pose serving as the ground truth.

We use cross-entropy as the supervision signal to update the model. Additionally, to prevent the model from forgetting the knowledge acquired during the training phase, we gradually decrease the learning rate as the number of test samples increases.

## 4 Experimental Results

### 4.1 Experiment Setting

**Implementation Details.** We use SAPIEN [27] and the PartNet-Mobility [28] dataset to set up the experiment environment. We employ a Franka Panda on-the-fly suction gripper to execute the end-effector actions. We randomly sample about 12K successful manipulation samples across 20 categories to build dataset $D_o$. After that, we augment $D_o$ by supplementing each example with the necessary training information, such as mask, joint direction, joint type, etc., thus obtaining $D_{aug}$. For testing, we sample about 1K successful manipulation samples across 30 categories to ensure the objects can be manipulated. Follow the work before [26], we only evaluate the pulling action primitives. We finetuned LLaMA-Adapter [29] on an 80G A800 GPU for 10 epochs. Each epoch takes about 1 hour. Further details can be found in Appendix B.4, B.3.

**Evaluation Metrics.** We use manipulation success rate to measure performance, which is the ratio of successful samples to total test samples. For the definition of success in pulling, we require

Table 1: Comparisons with baseline methods and ablation study.

| Method | Train Categories | | | | | | | | | | | | | | | |
|---|---|---|---|---|---|---|---|---|---|---|---|---|---|---|---|---|
| UMPNet [24] | 0.23 | 0.36 | 0.41 | 0.22 | 0.24 | 0.30 | 0.43 | 0.34 | 0.51 | 0.21 | 0.66 | 0.27 | 0.23 | 0.23 | 0.29 | 0.60 |
| FlowBot3D [25] | 0.45 | 0.48 | 0.45 | 0.32 | 0.32 | 0.37 | 0.43 | 0.23 | 0.26 | 0.14 | 0.39 | 0.31 | 0.38 | 0.32 | 0.23 | 0.43 |
| ManipLLM [26] | 0.72 | 0.56 | 0.32 | 0.79 | 0.48 | 0.53 | 0.66 | 0.69 | 0.39 | 0.52 | 0.53 | 0.4 | 0.64 | 0.73 | **0.62** | 0.52 |
| base model | 0.21 | 0.51 | 0.25 | 0.52 | 0.10 | 0.40 | 0.30 | 0.31 | 0.50 | 0.29 | **1.00** | 0.79 | 0.13 | 0.56 | 0.23 | 0.80 |
| Ours-w/o pretrain | 0.29 | 0.68 | 0.33 | 0.67 | **0.95** | 0.60 | 0.68 | 0.46 | 0.50 | 0.54 | **1.00** | **0.83** | 0.28 | 0.72 | 0.31 | 0.93 |
| Ours-w/o pos | 0.81 | 0.88 | 0.42 | 0.86 | 0.86 | 0.40 | 0.92 | 0.73 | 0.63 | 0.79 | **1.00** | 0.76 | 0.81 | 0.93 | 0.31 | **1.00** |
| Ours-w/o rot | 0.83 | 0.83 | 0.33 | 0.83 | **0.95** | 0.40 | 0.95 | 0.81 | 0.69 | 0.71 | **1.00** | 0.76 | 0.49 | 0.90 | 0.31 | **1.00** |
| Ours-w/o tta | 0.83 | **0.90** | 0.58 | **0.90** | 0.86 | 0.53 | 0.95 | 0.77 | **0.75** | 0.82 | **1.00** | 0.78 | 0.83 | 0.93 | 0.31 | **1.00** |
| Ours | **0.90** | 0.88 | 0.58 | 0.84 | 0.81 | 0.40 | **0.97** | 0.81 | 0.75 | 0.79 | **1.00** | 0.76 | **0.85** | **0.94** | 0.31 | **1.00** |

| Method | Train Categories | | | | | Test Categories | | | | | | | | | | |
|---|---|---|---|---|---|---|---|---|---|---|---|---|---|---|---|---|
| | | | | | AVG | | | | | | | | | | | AVG |
| UMPNet [24] | 0.32 | 0.30 | 0.11 | 0.58 | 0.34 | 0.36 | 0.36 | 0.38 | 0.47 | 0.21 | 0.12 | 0.24 | 0.23 | 0.28 | 0.12 | 0.28 |
| FlowBot3D [25] | 0.19 | 0.33 | 0.23 | 0.47 | 0.33 | 0.29 | 0.47 | 0.64 | 0.31 | 0.27 | 0.30 | 0.09 | 0.41 | 0.35 | 0.37 | 0.35 |
| ManipLLM [26] | **0.39** | **0.75** | 0.44 | 0.67 | 0.57 | 0.32 | 0.22 | 0.65 | 0.69 | 0.38 | **0.85** | 0.27 | 0.53 | 0.26 | 0.38 | 0.47 |
| base model | 0.00 | 0.42 | 0.13 | 0.60 | 0.40 | 0.09 | 0.10 | 0.68 | 0.17 | 0.50 | 0.56 | 0.13 | 0.27 | 0.04 | 0.38 | 0.34 |
| Ours-w/o pretrain | 0.20 | 0.52 | 0.18 | **1.00** | 0.56 | 0.27 | 0.09 | 0.73 | **0.83** | 0.43 | 0.64 | 0.22 | 0.63 | 0.11 | 0.81 | 0.44 |
| Ours-w/o pos | 0.00 | 0.73 | 0.65 | 0.60 | 0.77 | 0.36 | 0.63 | 0.74 | 0.33 | 0.79 | 0.78 | 0.74 | 0.76 | 0.71 | **0.88** | 0.71 |
| Ours-w/o rot | 0.00 | 0.67 | 0.31 | 0.60 | 0.71 | 0.36 | 0.14 | 0.63 | 0.33 | **0.86** | 0.63 | 0.61 | 0.68 | 0.14 | 0.50 | 0.45 |
| Ours-w/o tta | 0.00 | 0.72 | **0.68** | 0.60 | **0.80** | 0.36 | **0.70** | 0.74 | 0.50 | **0.86** | 0.82 | **0.83** | **0.85** | 0.73 | **0.88** | **0.76** |
| Ours | 0.20 | 0.72 | **0.68** | 0.80 | **0.80** | **0.45** | 0.65 | **0.76** | 0.50 | **0.86** | 0.81 | 0.78 | 0.83 | **0.77** | **0.88** | 0.75 |

a difference of more than 0.01 units between the initial and final object poses or 0.5 relative to the total motion range of the articulated part, and we also require the dot product of the predicted gripper direction and the actual movement direction of the object to be greater than 0.3.

**Baseline & Ablation Setting.** (1) UMPNet, FlowBot3D: two expert suction manipulation models that use deep learning methods to utilize visual perception information for predicting the SE(3) pose to manipulate the articulated object. (2) ManipLLM: the state-of-the-art MLLM model for predicting the SE(3) pose. (3) Base model: our base model trained solely on $D_o$ without any other chain-of-thought data like ManipLLM. (4) Ours-w/o pretrain: change our model to base model and do not use TTA module. (5) Ours-w/o pos, Ours-w/o rot: the former doesn't use the position correction module, and the latter doesn't use the rotation correction module. Both of them do not use TTA module. (6) Ours-w/o tta: Do not use the TTA module when testing. (7) Ours: using four different types of data to train the MLLM to understand the prompt and applying AIC MLLM framework. For the settings from (4) to (7), we perform four corrections.

### 4.2 Main Results & Analysis

**AIC MLLM can correct itself based on erroneous experiences, thereby achieving improved performance.** Tab. 1 summarizes all the experimental results. From the results, it is evident that our model shows significant improvement over the base model in both train and test categories. Moreover, As the number of corrections increases, the success rate progressively improves (Fig. 4). This indicates that our model can correct itself based on failed experiences and continuously gather information and reflect on it to make further corrections as these experiences accumulate.

**AIC MLLM is more generalizable.** Our model achieved a 0.75 success rate on unseen test categories, significantly outperforming the best baseline (Tab. 1). This demonstrates the strong generalization capability of our framework, indicating that the information extracted pertains to the universal characteristics of articulated objects.

**Training the model to understand the prompt is important.** In Tab. 1, by comparing Ours-w/o pretrain and Ours-w/o TTA, we can see that training the model to understand interface prompts is crucial. This allows our model to improve from 56% to 80% on the train categories and from 44% to 76% on the test categories.

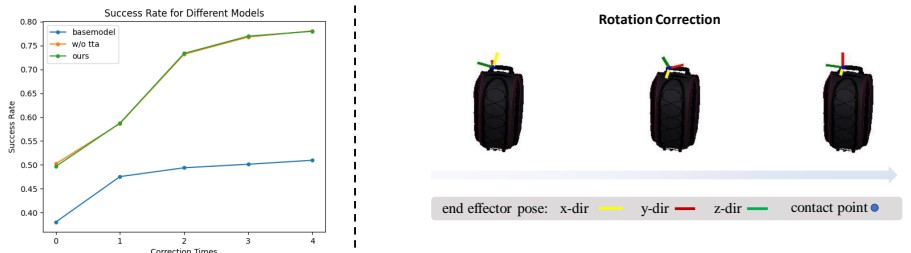

Figure 4: **Ablation of the proposed method and visualization.** The image on the left illustrates the correlation between success rate and correction times. On the right, the correction process is depicted with the aid of simulation.

**Position correction & rotation correction are both important.** By comparing Ours-w/o pos, Ours-w/o rot, and Ours-w/o tta, we observe that using only rotation correction or only position correction results in a performance drop. This demonstrates that both rotation and position corrections are essential, and their combined effect is crucial for our model to achieve optimal performance.

**TTA is useful for AIC MLLM to adapt to the test domain.** From the table 1, we can observe that comparing ours-w/o TTA with ours, the addition of the TTA module resulted in higher or unchanged success rates in 6 unseen test categories and 15 seen train categories. This indicates that incorporating TTA allows the model to adapt to a broader range of object categories. A slight amount of forgetting occurred in a few categories, which may be due to the small number of samples in those categories. However, the minimal difference in average success rates between the two shows that our TTA strategy effectively mitigates forgetting.

### 4.3 Real-world Experiment

After validating the effectiveness of our method in simulation, we conduct experiments in the real world. The detailed experimental setup and results can be found in Appendix F.

## 5 Conclusion

We introduce AIC MLLM, a framework leveraging MLLM to correct SE(3) position predictions through learning from low-level interaction failures. We design visual and textual prompts for guiding position and rotation corrections, along with the integration of a feedback information extraction module to adaptively correct pose predictions based on identified failure causes. We implement the test-time adaptation module to improve manipulation stability.Comprehensive experiments showcase the effectiveness of AIC MLLM across simulated and real-world environments.

## 6 Limitation & Future Work

In this work, we primarily focus on low-level contact pose correction for cross-category articulated object manipulation. However, several potential limitations can be considered and addressed in future work. Firstly, while our current approach focuses on pose-level corrections, existing frameworks like REFLECT [6] handle only high-level corrections. Future work could integrate these frameworks to achieve long-term corrections at both the pose and high levels. Secondly, our task involves correcting action primitive poses rather than operating in a closed-loop manner. Leveraging more temporal data and designing alternative prompts could offer promising solutions for improvement. Additionally, our framework utilizes the off-the-shelf foundation model VIP-LLaVA and SAM, which is not specifically designed for object manipulation property reasoning. Fine-tuning this model on object-centric manipulation data could enhance the overall performance of our framework.

**Acknowledgments**

We would like to thank reviewers for their valuable comments and feedback. This work was supported by the National Youth Talent Support Program (8200800081) and National Natural Science Foundation of China (No. 62136001).

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

# Appendix

## A  More related works

**Multimodal Large Language Models (MLLM).** Large language models (LLM) have impressive power in natural language processing tasks. Consequently, some researchers aim to develop even more powerful MLLMs based on LLMs to tackle a broader range of vision-language tasks [30, 31, 12, 32, 29, 13]. CLIP [32] first established a connection between language and vision. The BLIP series [30, 31, 12] freezes the LLM and the image encoder, training their bridge, Q-Former, to align the two modalities. It also employs instruction tuning to enhance its ability to follow instructions. LLaVA [13] uses a simple fully connected layer to build the bridge of the LLM and the image encoder. LLaMA-Adapter [29] utilizes projection and adapters to make the model have the multimodal ability and reduce training costs. Additionally, closed-source MLLMs like GPT-4V [33] are more powerful than the open-source models we mentioned previously.

**Robotic Manipulation.** Reinforcement learning [34, 35, 36, 37], imitation learning [38, 39], and deep learning for visual understanding [40, 25, 41, 24] have been extensively applied in robotic manipulation. Recent studies like where2act [40] and Flowbot3d [25] combine prior knowledge with deep learning. Where2act [40] leverages point cloud data as input to initially predict the score of each point, followed by rotation prediction. UMPNet [24] employs a position network for position inference, then utilizes a action sampler to sample some directions of this position, and subsequently employs two networks to evaluate these directions. Flowbot3D [25] introduces an articulation flow to represent the point-wise potential motion and uses this representation to guide the manipulation. Flowbot++ [41] integrates joint axis modeling into Flowbot3D to make the trajectory more smooth. Following the emergence of powerful large language models, researchers have started employing them to address the generalization challenges in manipulation tasks. Our work also focuses on leveraging large-scale models to enhance the robustness of algorithm in manipulation tasks.

**Robotic Manipulation with MLLMs.** MLLMs have also been applied to robotic manipulation [42, 43, 44, 45, 26], with models such as Palm-e [42], Voxposer [43], RT-2 [45], and ManipLLM [26] showing promise for high-level planning, mid-level cost function injection, and direct low-level pose output. While high-level correction has been widely studied, low-level correction remains largely unexplored.

**MLLMs with Prompt Instruction.** The performance of MLLMs heavily depends on input prompts. With the integration of diverse input modalities, prompt instructions have become essential for MLLMs to perform specific tasks [46, 21]. Some research focus on image labeling and annotating coordinates as prompts [47, 48]. VIP-LLaVA [21] and SPHINX-V [46] have been trained to understand visual prompts. In this paper, we use visual prompts and textual prompts to indicate unmovable parts and potential directions for manipulation.

## B  Detailed about Methodology and Implementation

### B.1  Noise Injection in Rotation-Related Interface Data

Note that, in simulator, we can obtain the accurate joint type, joint axis direction and norm direction, and calculate the actual movement direction of the part. However, in real-world scenarios, the estimated joint axis direction may not be accurate and can be noisy. To mitigate error accumulation, we introduce noise into the training process by adding a random rotation between -20° and 20° around the obtained axis direction. This approach enhances the robustness of the model's predictions based on these priors.

## B.2 Rotation-related Information Extraction

In this section, we will provide a detailed introduction to how we perform joint type classification. After the end effector (EE) reaches the predicted pose and adheres to the object, it will move along the predicted gripper direction for a certain distance. We will record the pose trajectory of EE for 20 frames during this process. We denote the i-th pose in the trajectory as $^{\mathrm{W}}P_i^E$. We perform the operation of subtracting the previous pose from the current pose for all poses in the sequence which can be denoted as $\vec{v} = {}^{\mathrm{W}}P_i^E - {}^{\mathrm{W}}P_{i-1}^E$. We gather the $\vec{v}$ to form a vector list. Then we calculate the angle between adjacent vectors. Next, we set a threshold. If all the angles are less than this threshold, then the joint will be considered a prismatic joint; otherwise, it will be considered a revolute joint.

## B.3 Model Architecture Details

Our model architecture follows LLaMA-Adapter [29]. Given an RGB image $I \in \mathbb{R}^{H \times W \times 3}$, we employ pre-trained CLIP [32] as the visual encoder to extract visual features. Text prompts $P$ are transformed into text features using the tokenizer from the pre-trained 7B LLaMA [49]. After aligning the visual and text feature representations via a multi-modal projection module consisting of 32 transformer blocks, LLaMA acts as the decoder to output appropriate answers. The model adopts a set of learnable adaptation prompts (LoRa [50]) and inserts these prompts into the multi-modal alignment module and LLaMA for quick finetuning to other tasks. Since we used the LLaMA Adapter for pretraining, we adopt the same network structure. This is because its alignment layer, already trained on a large amount of real-world data, can perfectly align the feature of visual encoder CLIP with LLaMA, providing strong generalization for downstream tasks such as robotics. During finetuning, we use the pretrained model of LLaMA-Adapter, and only finetune the injected LoRa to adapt to the tasks presented in Figure 2. This approach ensures that the model retains most of its general reasoning abilities for real-world applications while gaining additional robotics-related correction capabilities.

## B.4 More Implementation Details

In this section, we will include additional implementation details that were not mentioned in the main paper. During data collection for the dataset $D_o$, we randomly sample the contact points and the gripper directions, then determine if the interaction is successful. If it is, we save the data; otherwise, we delete it. We sample both pulling and pushing action primitives. During the training phase, for the VQA task 'Mask Position Reasoning,' we mentioned that we randomly select N points. In implementation, N is set to 20. During testing, it is important to note that all MLLM temperatures are set to 0. In the tta training process, we initialize the learning rate (lr) to 5e-8 and set the weight decay to 2e-3. Every 300 iterations, we reduce the lr by 70%.

Table 2: Ablation study on mobility detection

| Method | Train Categories | | | | | | | | | | | | | | | |
|---|---|---|---|---|---|---|---|---|---|---|---|---|---|---|---|---|
| VIP-LLaVA | 0.83 | 0.90 | 0.58 | 0.90 | 0.86 | 0.53 | 0.95 | 0.77 | 0.75 | 0.82 | 1.00 | 0.78 | 0.83 | 0.93 | 0.31 | 1.00 |
| GT | 0.86 | 0.93 | 0.50 | 0.90 | 0.86 | 0.40 | 0.92 | 0.77 | 0.88 | 0.79 | 1.00 | 0.78 | 0.83 | 0.93 | 0.46 | 1.00 |

| Method | Train Categories | | | | AVG | Test Categories | | | | | | | | | | AVG |
|---|---|---|---|---|---|---|---|---|---|---|---|---|---|---|---|---|
| VIP-LLaVA | 0.00 | 0.72 | 0.68 | 0.60 | 0.80 | 0.36 | 0.70 | 0.74 | 0.50 | 0.86 | 0.82 | 0.83 | 0.85 | 0.73 | 0.88 | 0.76 |
| GT | 0.00 | 0.75 | 0.69 | 0.60 | 0.80 | 0.45 | 0.65 | 0.77 | 0.50 | 0.79 | 0.79 | 0.78 | 0.80 | 0.74 | 0.81 | 0.74 |

# C  More Ablation

Tab. 2 presents the ablation study results comparing the VIP-LLaVA method and the GT method. VIP-LLaVA uses the VIP-LLaVA model to predict part mobility, while GT uses the ground truth interaction map. The table shows that the results from VIP-LLaVA are quite similar to those from GT in this simulation experiment. This indicates that our method is not sensitive to the mask in this simulation environment, demonstrating a certain level of robustness.

# D  Example of Prompts

## D.1  Position correction prompt

Firstly, we make use of the mask information to help the MLLM get a more precision prediction by the following prompts.

| Is there a red mask in the image? |
| --- |
| **AIC-MLLM:** No. |
| Is there a red mask in the image? |
| **AIC-MLLM:** Yes. |

| Determine if following point is in the red mask: (172,47),(117,192),(323,251),(195,9),(211,277),(242,292),(87,70),(88,314),(193,39),(87,174),(88,165), (25,333),(72,265),(115,243),(197,335),(99,177),(243,285),(147,147),(288,265),(185,127) |
| --- |
| **AIC-MLLM:** No,Yes,No,No,No,No,No,No,No,No,No,No,Yes,No,No,No,Yes,No,Yes |
| Determine if following point is in the red mask: (11,104),(212,138),(182,125),(156,111),(258,27),(217,151),(309,307),(174,148),(29,67),(35,295),(73,297), (302,218),(259,287),(265,27),(199,61),(44,290),(88,33),(133,232),(255,36),(256,290) |
| **AIC-MLLM:** No,No,Yes,Yes,Yes,No,No,Yes,No,No,No,No,Yes,No,No,No,Yes,No,No |

For position correction using the red mask image, we reassess the validity of the previously predicted contact point. If the contact point is deemed ideal, the MLLM will confirm this by replying "Yes." Otherwise, it will reply "No."

| Given the part with a red mask known to be unmovable in the image, is the (112,171) considered as the successful contact point? |
| --- |
| **AIC-MLLM:** Yes. |
| Given the part with a red mask known to be unmovable in the image, is the (160,160) considered as the successful contact point? |
| **AIC-MLLM:** No. |

Then, MLLM makes the prediction using the image with red mask.

| Specify the successful contact point and orientation of pulling the object. DONOT predict the successful contact point in the red mask part. |
| --- |
| **AIC-MLLM:** The contact point is at (170, 179), the gripper direction is [48, -1, -12], the gripper forward direction is [-1, 1, 49]. |
| Specify the successful contact point and orientation of pulling the object. DONOT predict the successful contact point in the red mask part. |
| **AIC-MLLM:** The contact point is at (143, 174), the gripper direction is [48, 1, 12], the gripper forward direction is [1, -49, -10]. |

MLLM assesses whether the predicted point is a successful point.

| You pull the object in the picture by point(170,162),gripper direction[47,11,-11],gripper forward direction[10,-49,1],if it will success answer only Yes , else answer only No. |
| --- |
| **AIC-MLLM:** No. |
| You pull the object in the picture by point(179,179),gripper direction[38,-28,13],gripper forward direction[13,45,-19],if it will success answer only Yes , else answer only No. |
| **AIC-MLLM:** Yes. |

The FIE acquires environmental information, including joint type, joint axis and the normal direction of previously predicted contact point. The MLLM uses this information, along with the contact point, to infer whether to correct its response by suggesting a new gripper direction.

> You're trying to pull an object part, the joint's type is REVOLUTE, the joint axis direction is [-22, -1, 44], Your contact point is (79, 170), the norm direction of the point is [-47, 9, 12]. Please tell me your gripper direction.

**AIC-MLLM:** The gripper direction is [-47, 13, -13].

> You're trying to pull an object part, the joint's type is REVOLUTE, the joint axis direction is [10, 6, -48], Your contact point is (170, 159), the norm direction of the point is [-29, 0, 40].Please tell me your gripper direction.

**AIC-MLLM:** The gripper direction is [-28, 35, -20].

# E   Example of Failed Case and correction process

We present some examples of corrections from our simulation experiments. First, for position correction in Figure 5, we used red dots to represent the points predicted by the model and showed the effects before and after correction. Then, we demonstrated the model's ground truth movable mask. Next is the rotation correction shown in Figure 6, where we used a coordinate system to represent the direction of the end effector. The rightmost image in Figure 1 shows how this coordinate system is defined, with the red axis indicating the direction of the end effector's pulling motion. The first object is a display from PartNet-Mobility that requires rotating the screen, a revolute joint case. The second and third objects are doors that need to be opened, also involving revolute joints.

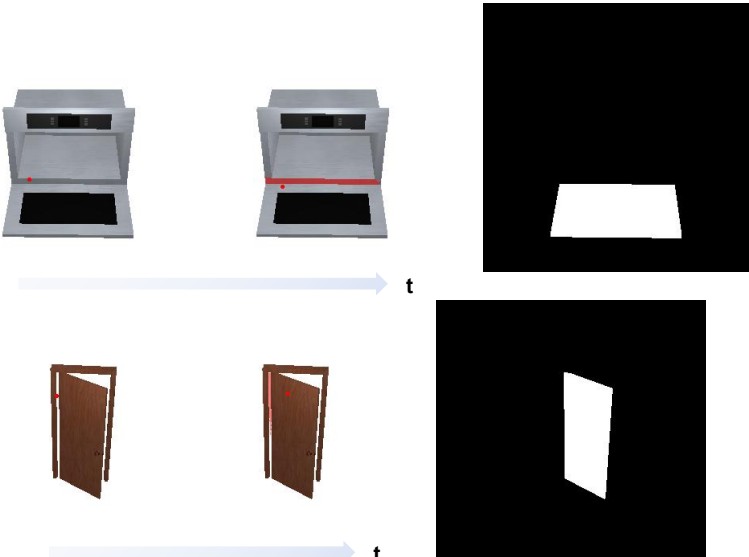

Figure 5: **Examples of position correction.** The first figure is the original prediction of the contact point, and the red dot in the second figure is a new prediction of contact point keeping away from the red mask. And the third figure is the interaction map where white area is the movable part of the object.

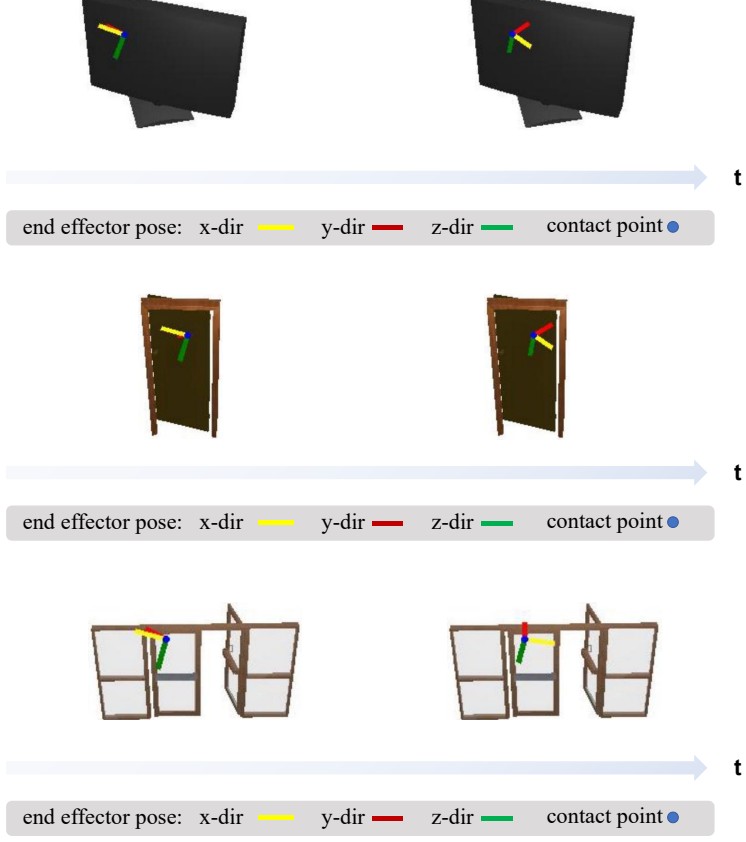

Figure 6: **Rotation correction**. Correction times increase sequentially along the timeline.

# F    Real-world experiments

To validate our end-to-end method beyond simulations, we conducted real-world experiments using a Franka Emika robotic arm equipped with an Intel RealSense D415 sensor without finetuning in real-world.

**Experimental Setup.** The setup included a Franka Emika Panda robotic arm, known for its precision and versatility. An Intel RealSense D415 sensor was used to capture RGB-D data, providing 3D information necessary for the robotic arm's operations. We closed the traditional gripper and applied double-sided tape to its head as a suction gripper. We tested 8 different articulated objects. Specifically, we chose 5 revolute objects: Toilet, Trashcan, Microwave, Cabinet and Faucet, and we chose 3 prismatic objects: Pumpkin-like Mug, Pot and Drawer. Each object undergoes five trials with random camera distances and view angles. Each trial allows up to five attempts. If the method succeeds at least once within these five attempts, it is considered successful; otherwise, it is deemed a failure. For this statistical experiment, we visualize the model's predictions and control the robot accordingly. And we do not perform TTA module. For each object, we provided a detailed description of each task involving that object. For the toilet, the robot needs to lift the toilet lid. For the trashcan, it needs to open the trashcan. When working with the microwave, the robot is required to open the microwave door. For the pumpkin-like mug, the task involves removing its lid. For the cabinet, the robot needs to open its door, and for the pot, it must lift the lid. Additionally, the robot is required to adjust the direction of the faucet and pull the drawer.

**Initialization.** The robotic arm and the D415 sensor were calibrated to ensure accurate spatial data capture and precise movements.

**Image Processing.** The D415 sensor captures high-resolution RGB-D images, which are processed to meet the requirements of our pipeline. Specifically, our method necessitates images sized at 336x336 pixels. To preserve information during resizing, we apply both cropping and padding techniques. This involves cropping or adding white borders around the captured images to resize them to 336x336 pixels. The resulting image, whether cropped or padded, is then used for further analysis in our end-to-end pipeline.

**Task Execution.** Using our end-to-end method, the robotic arm was directly controlled to perform tasks based on the RGB-D data input. The tasks involved the arm autonomously approaching and manipulating objects within the workspace.

**Demonstration.** The robotic arm demonstrated its ability to perform these tasks seamlessly, showcasing the practical applicability and robustness of our end-to-end method in handling real-world scenarios. In the video, the robot was tasked with correctly pulling the cabinet door open. As demonstrated, the robot failed on its first attempt. However, after applying our framework for correction, it successfully opened the cabinet door.

**Experimental Result.** Table 3 presents the real-world performance results. Our framework demonstrates an improvement in the average success rate from 37.5% to 65%, reflecting a trend similar to that observed in the simulation experiments. This consistency indicates that our framework effectively transfers from simulation to real-world applications.

Table 3: Real-world performance. The table shows the success rates for our framework across various objects and the average performance.

| Method | Toilet | Trashcan | Microwave | Mug | Cabinet | Pot | Faucet | Drawer | **AVG** |
|---|---|---|---|---|---|---|---|---|---|
| w/o correct | 3/5 | 1/5 | 2/5 | 1/5 | 3/5 | 3/5 | 0/5 | 2/5 | 37.5% |
| w/ correct | 5/5 | 3/5 | 3/5 | 3/5 | 4/5 | 3/5 | 1/5 | 4/5 | 65% |

