# OpenReview forum: "Autonomous Interactive Correction MLLM for Robust Robotic Manipulation"
_robot-learning.org/CoRL/2024/Conference — CoRL 2024_

### Official Review · Reviewer_q3ru · 2024-07-20
**Weak Reject (Concerns about Scope, Limited Hardware Experiments)**

**Originality:** 3
**Technical Quality:** 3
**Clarity Of Presentation:** 2
**Potential Impact:** 2
**Recommendation:** 2
**Confidence:** 3

**Review:**

Strengths:
- The paper is well-written, straightforward, and the provided figures and diagrams are great. The provided supplementary video is also quite helpful for understanding the framework.
- It is good to see some initial demonstrations on a real-world setting
- The related work is pretty thorough and baselines are well-motivated. I appreciate that the authors compared against several ablations of their method to understand the different design choices.

Weaknesses:
- Where is the limitations section?
- The proposed idea of iteratively masking out regions in which a manipulation action failed, before re-prompting a VLM to identify new interaction point candidates seems like it is a bit overkill. I think the main challenge with detecting failures/successes in manipulation is not really bottlenecked so much by perception/object identification -- recent works like RoboPoint (which should be cited btw), along with several other works that use VLMs for planning have shown that with the right combination of open-vocabulary object detectors can generally localize the kinds of semantic parts this paper looks at
- For example, in the real-world demonstration, why is the initial point of the cabinet handle so poorly predicted (it is on the joint)? This seems like current SOTA open-vocabulary object detectors should really be able to localize this with a lot more accuracy from the beginning -- is this because LLaMa just really struggles with keypoint prediction?
- Much of the prompting seems very task-specific, and relying privileged information like knowing whether a cabinet is revolute/prismatic, etc. It is unclear how much prompt tuning will be required to use this approach on other tasks with their own specifics.
- Still, I think that this approach is reasonable, but the bigger issue is that it seems to view failure recovery as just re-predicting a new pose for the robot to go to. I think this may apply to some simple grasping/cabinet opening related tasks, but is generally not applicable to any kind of more precise/fine-grained task that requires closed loop control. Asking a VLM to reason about 6DoF actions is also a really tough ask, and I wonder if a more principled approach could be more useful (i.e. detecting grasp candidates either analytically or from a grasp generator like 6DoF grasp or AnyGrasp or ContactGraspNet, etc. and having a VLM choose from these seems like a better posed problem).

**Quality Of The Limitations Section:**

1

**Questions For Rebuttal:**

- Have you tried other methods of keypoint detection beyond just prompting LLaMa? Can you ablate a few in the rebuttal (maybe RoboPoint, GroundedSAM / RAM, etc.)?
- How do you see this approach scaling to more complex tasks where failures are not at the pose-level, but may be either even more precise low-level failures, or higher-level task-planning failures?
- How sensitive is this approach to the prompts? Did you have to do a lot of prompt engineering, and is it possible to do more ablations on how much this matters?
- In practice, how will you retrieve information like the joint type of an articulated object? This seems like a big assumption, and so it would be useful to either show how this works on a different task, or show how you can sense these things possibly from vision, etc.

**Robotics Focus:**

2

**Summary Of Paper:**

This work proposes a method for failure detection and recovery using the few-shot reasoning capabilities of foundation models. The proposed system is evaluated on simulated benchmarks along with a real-world demonstration on the Franka.

**Summary Of Recommendation:**

I appreciate the authors efforts to present their approach clearly and to demonstrate it on both simulated and real scenarios; the main concern is the amount of task-specific assumptions in this approach, and how it will really scale to a variety of real-world tasks.

---

### Official Review · Reviewer_ygE4 · 2024-07-20
**Autonomous Interactive Correction MLLM for Robust Robotic Manipulation**

**Originality:** 3
**Technical Quality:** 2
**Clarity Of Presentation:** 3
**Potential Impact:** 2
**Recommendation:** 3
**Confidence:** 3

**Review:**

Strengths:
- The paper is well written and easy to follow. + the figures are clear and helpful.
- The method of correcting failures by prompting a model with masks and language is clever
- The training curriculum designed for progressively shaping the MLLM’s understanding of the desired prompts is interesting
- Experimental results indicate that this is a promising direction
- The proposed approach is demonstrated on a real-world robot

 Weaknesses:
- The approach seems heavily tailored toward certain types of failures (grasping immovable parts of articulated objects). I would have liked to see more discussion about the limitations and how the framework might be extended for other types of failures.
- While the training curriculum is well described, there are very little details or discussion about the underlying model being trained.
- There is very little details or discussion about the real-world robot experiments. It’s not clear what tasks were demonstrated in the real-world and the supplementary video only shows one task (opening a cupboard door) and only shows one failed execution / recovery. I would have liked to see much more details about the tasks performed and results of the real-world experiments because I am left with questions about how well the promising performance from the simulated benchmarks transfers to the real-world.
- The feedback information extraction is limited by the performance of some off-the-shelf models (e.g. VIP-LLaVA and SAM)

Minor suggestions:
- Typo on line 103: Postional -> Positional
- On line 149 it’s not clear to me what ‘process’ in this sentence is supposed to mean: “We introduce the process of our framework during the testing phase.”
- On line 258, missing space between Results &

**Quality Of The Limitations Section:**

1

**Questions For Rebuttal:**

- While the training curriculum is clearly described, the actual model and architecture are not immediately clear to me. At the beginning of your method section you start by saying “we enable the model to predict poses”, and throughout the method section you very frequently reference the model, but as far as I can tell you do not define anything about the model. In the experiment phase you do specify fine tuning LLaMA-Adapter, but I feel like the architecture of the model could be better described in the context of your method.
- In section 3.3.1 you describe your method of extracting joint type and axis based on interaction attempts… e.g. if the motion trajectory is an arc it is classified as revolute. But how are the initial trajectories generated? From what is described in the overview (section 3.1) the model is predicting poses, and per the description of the testing phase (3.3) the model is outputting a 2D contact point (lifted into 3D by the depth map) and rotation. But it’s not clear to me how is the post contact motion trajectory generated?
    - It might be worth directly describing the problem formulation up front to make it really clear the domain, action space, etc that your framework is operating in.
- Does the fifth step of position correction help? You ask the model to evaluate whether the newly predicted pose will result in a successful interaction — but is that evaluation used for anything? Did you perform experiments without that step?
- I appreciate the included details about LLaMA-Adapter fine tuning. I am curious how some of the design choices were made. E.g. decision of visual encoder, number of transformer layers for alignment bridge, etc. Did you experiment with different options or did you have some a priori evidence that informed these decisions?
- What tasks were performed in the real-world with the Franka robot? How does performance in the real-world compare to the performance you report on the simulated benchmarks?
- How might the framework be extended for other types of failures and tasks beyond grasping and pulling parts of articulated objects?

**Robotics Focus:**

4

**Summary Of Paper:**

The authors propose fine tuning an off-the-shelf multimodal language model to make use of low-level interaction experiences and correct pose predictions. The model can be prompted by either a visual mask highlighting parts for position correction or language instructions describing the desired correction. The training curriculum is carefully designed to shape the models understanding of the desired prompts. Additionally, the authors propose a test time adaption strategy that updates the model from online experience.  Experiments are performed in simulation with SAPIEN and PartNet-Mobility. Finally, the failure correction approach is demonstrated in the real world using a Franka robot arm to open a cupboard door.

**Summary Of Recommendation:**

I feel that the paper is missing some details that I expect to be addressed during the rebuttal period. But the overall approach seems interesting and worth sharing with the robotics learning community.

---

### Official Review · Reviewer_azV8 · 2024-07-21
**Autonomous Interactive Correction MLLM for Robust Robotic Manipulation: Recommend to accept with significant revision**

**Originality:** 3
**Technical Quality:** 3
**Clarity Of Presentation:** 2
**Potential Impact:** 3
**Recommendation:** 3
**Confidence:** 4

**Review:**

The topic of self-improvement is very important to robot learning, and the method presented here is novel and interesting. The claims of the paper could be more clearly articulated and supported. Revision could strengthen the paper by better clarifying the scope of what the authors have (and haven’t!) investigated, articulating limitations of the current work and next steps to identify the limits of the method (assuming the method has not been pushed to its limits yet), and discussing experiments, including hardware experiments, with technical/scientific rigor. More specifically:

*Unclear task scope*. Regarding scope of tasks, the manuscript doesn’t have a clear message and seems to conflict with itself.
* The paper opens with emphasis on generalization.
* Method includes prismatic and revolute joints, claiming “For articulated objects, there are mainly two types of joints: prismatic and revolute.”
* Image in figure2 is a pot, where the main interaction I see would be to remove the lid–neither prismatic nor revolute.
* In evaluation, the manuscript states, “we only evaluate the pulling action primitives”.

A consistent story would include (a) what task categories would be covered in a hypothetical ideal case, (b) what is covered by this work, and (c) what motivated this choice of coverage.

*Missing limitations and next steps*. The method seems to have promise, but it’s not clear from the manuscript how well the authors have actually developed and tested it so far. It’s also not clear how, if at all, it has been integrated into a policy. A discussion of limitations and next steps would help the reader understand what the authors have actually investigated, and where the authors see potential from their work so far.

*Integration*: I don’t see an example of how the method is integrated into a policy, e.g. that will behave one way in success cases, and then in case of failure gracefully leverage the presented method to improve performance. However, perhaps the authors have only built the method to distill a set of failed experiences into a useful improvement, but not yet integrated it into a policy? Does the method as demonstrated so far require a human-selected series of failed episodes, or has the method been tested with a robot in a “self-play” mode where, starting from the first episode, a policy captures and categorizes successes and failures and uses them according to the method? *How is the intended task communicated to the robot in the first place? Or is the goal of the method only to discover possible affordances of articulated objects?*

*Masking*: The method trains on a red mask added to objects. How does this impact the model’s performance working with (identifying, segmenting, analyzing) objects that are already red?

**Clarification nits:**

~Line 200: “In the third step, we prompt the model to determine whether the previously predicted points are on unmovable parts, compelling it to reflect on the reason behind the previous failures…” What fraction of failures are actually due to incorrect points–are there other common failure modes?  If so, how does this method respond in such cases?

Line 159-161: not sure I understand what the authors are trying to communicate–consider a rewrite with clarity in mind.


*End-effector pose labels in figures*: Fig4: clarify the end-effector pose labels in annotation? The rightmost image shows left-dir pointing to the right, up-dir pointing into or away from the page, and I don’t see for-dir annotated in any of the images.

This notation is used again in the Appendix fig2, and there it is not clear what the intended success would look like for each task (e.g. the first object looks like a TV–not sure how it’s supposed to articulate; last object looks like a suitcase on wheels–is the goal to extend a handle, or push/pull the suitcase in a particular direction?), or how to map the “end effector pose” color labels to progress. The labels appear to set a coordinate space, but not to set a direction of intended movement, however the paper claims to improve the movement trajectory, not just the identification of a coordinate space.

*Polish*: Please check for spelling mistakes, there are at least a few typos per page, including some incorrect words.

*Originality*: I’m between “Good” and “Very Good”; better understanding the scope of the claims could clarify.

**Quality Of The Limitations Section:**

1

**Questions For Rebuttal:**

Method sounds interesting/promising, but some claims are not clearly articulated or supported.

1) I have trouble understanding the extent of the claims. Clearly articulating scope, stating what is *and isn’t* included in the contributions, would add clarity. E.g.

a) How deployable is the method now? Has it been tested in isolation, with failed episodes selected by a human and provided to the method? Has it been tested integrated into a policy, where the model will behave one way in success cases, and then in case of failure gracefully leverage the presented method to improve performance?

b) What tasks have the authors considered? Tested? Which tasks are out of scope here?

c) Statistical analysis would add credibility to experimental results.

2) Critically missing: Experiments–almost no information is provided. There is reference to a video, which contains an image of a moving robot. Need clarity on what experiments have been run, with statistical analysis.

3) Critically missing: limitations, next steps.

**Robotics Focus:**

3

**Summary Of Paper:**

With the stated goal of enabling robots to learn from mistakes, the authors propose a method they call “Autonomous Interactive Correction” (AIC), which takes as inputs image and end-effector positions from a set of failed object-interactions, and leverages a specially fine-tuned multi-modal foundation model to extracts relevant information about the failures to enable a better pose prediction for the next attempt.

**Summary Of Recommendation:**

I recommend accepting the paper with significant revision. The topic of self-improvement is very important to robot learning, and the method presented here is novel and interesting. Revision could strengthen the paper by better clarifying the scope of what the authors have (and haven’t!) investigated, articulating limitations of the current work and next steps to identify the limits of the method (assuming the method has not been pushed to its limits yet), and discussing experiments, including hardware experiments, with technical/scientific rigor.

---

### Author Rebuttal · Authors · 2024-08-11

First, we would like to thank all reviewers for their valuable comments and thorough reviews! We are happy that reviewers find our work interesting and important for robotics.

We have addressed each reviewer's concerns individually, following their suggestions, we have also made several major changes to our paper :

- **Task scope** We clarify the scope of this work, specifically highlighting the focus on low-level pose correction for atomic tasks involving articulated objects.

- **Limitation and future work section** We added a section detailing our limitations and potential future work. This section also makes our task scope more clear

- **Real-world experiment** We add more real-world robot experiments with statistical analysis in appendix.

- **Icon for end-effector pose.** We revised the coordinate naming for clarity and updated Fig 1 for better alignment. We hope this makes it easier to understand.

- **More details about model architecture** We added the model architecture details in the revised appendix B.2.

- **Polish**  We corrected spelling errors and improved sentence clarity throughout the paper.

- **Improve experiment** We improved the simulation experiments following reviewers' suggestions

Thanks again to all reviewers for taking the time and effort to help improve the paper. All changes have been highlighted in blue in the revised version of the paper.

---

### Decision · Program_Chairs · 2024-09-04

**Decision:**

Accept

**Comment:**

Update: The paper presents a novel method that improves spatial understanding of foundation models for correcting robot policy executions. The authors responded well in the rebuttals to reviewer feedback. I am recommending an accept for this submission.

Original: The reviewers agree that the problem being studied by this paper is important and timely. However, they are concerned aabout the generalizabiliy of the proposed method and the paper's findings, as well as the lack of clarity on how the method can be best integrated for downstream applications. I ask the authors to carefully address all of the reviewers' concerns during the rebuttal period.